# Mechanism of lipid droplet formation by the yeast Sei1/Ldb16 Seipin complex

Yoel A. Klug [1], Justin C. Deme[1,2,5], Robin A. Corey[3,5], Mike F. Renne [1,5], Phillip J. Stansfeld [3,4], Susan M. Lea [1,2✉] & Pedro Carvalho [1✉]

Lipid droplets (LDs) are universal lipid storage organelles with a core of neutral lipids, such as triacylglycerols, surrounded by a phospholipid monolayer. This unique architecture is generated during LD biogenesis at endoplasmic reticulum (ER) sites marked by Seipin, a conserved membrane protein mutated in lipodystrophy. Here structural, biochemical and molecular dynamics simulation approaches reveal the mechanism of LD formation by the yeast Seipin Sei1 and its membrane partner Ldb16. We show that Sei1 luminal domain assembles a homooligomeric ring, which, in contrast to other Seipins, is unable to concentrate triacylglycerol. Instead, Sei1 positions Ldb16, which concentrates triacylglycerol within the Sei1 ring through critical hydroxyl residues. Triacylglycerol recruitment to the complex is further promoted by Sei1 transmembrane segments, which also control Ldb16 stability. Thus, we propose that LD assembly by the Sei1/Ldb16 complex, and likely other Seipins, requires sequential triacylglycerol-concentrating steps via distinct elements in the ER membrane and lumen.

[1] Sir William Dunn School of Pathology, University of Oxford, Oxford, UK. [2] Center for Structural Biology, Center for Cancer Research, National Cancer Institute, Frederick, MD, USA. [3] Department of Biochemistry, University of Oxford, Oxford, UK. [4] School of Life Sciences & Department of Chemistry, University of Warwick, Coventry, UK. [5]These authors contributed equally: Justin C. Deme, Robin A. Corey, Mike F. Renne. ✉email: susan.lea@nih.gov; pedro.carvalho@path.ox.ac.uk

L ipid droplets (LDs) are storage organelles for neutral lipids, such as triacylglycerol (TAG), with central roles in lipid and energy metabolism[1,2]. Despite their importance and links to metabolic diseases[3], the mechanisms controlling LD formation and dynamics remain poorly understood.

LDs display a unique architecture with a hydrophobic core composed of neutral lipids enclosed in a phospholipid monolayer[4,5] with a set of LD-specific proteins[6,7]. In recent years, various proteins and lipids have been implicated in LD biogenesis, and a general model of LD biogenesis has emerged[8,9]. Assembly of LDs occurs at the endoplasmic reticulum (ER)[10,11], and is dependent on synthesis of neutral lipids[12]. At low concentrations neutral lipids disperse within the acyl chain region of the ER membrane lipids, but as their concentration rises, they phase separate from surrounding phospholipids and coalesce into a lens-like structure between the leaflets of the ER bilayer[13–15]. As this neutral lipid lens grows, it gives rise to a nascent LD that buds from the ER enclosed in a phospholipid monolayer derived from the cytosolic leaflet of the ER bilayer[16].

The ER sites of LD biogenesis are marked by Seipin, an evolutionarily conserved ER membrane protein[17–21]. In humans, Seipin is encoded by the Bernadelli-Seip congenital lipodystrophy type 2 (BSCL2), a gene frequently mutated in familial forms of lipodystrophy[22]. At the cellular level, loss of Seipin results in aberrant LDs, which become tiny, often clustered, interspersed with a few supersized LDs[21,23,24], likely due to impaired maturation[17] and contacts with the ER[19,20]. Acute depletion of Seipin in human cells resulted in heterogenous LDs, demonstrating that Seipin is also required for the maintenance of mature LDs[25].

Seipin has two transmembrane (TM) segments proximal to the N- and C-termini separated by an extended luminal domain[26]. Cryogenic-electron microscopy (cryo-EM) structures of human and fly Seipin luminal domains showed homo-oligomeric ring assemblies of 11 and 12 subunits, respectively, an arrangement essential for function[27,28]. Despite the different subunit number within the Seipin rings, the individual Seipin protomers adopted a similar fold with 8-strand β-sandwich typical of certain lipid-binding proteins. While capable of binding anionic phospholipids in vitro, the in vivo relevance of these observations is still unclear[27]. Both in human and fly Seipin, the luminal β-sandwich is capped by a hydrophobic helix, which in the oligomer is positioned at the inner surface of the ring, protruding into the luminal leaflet of the ER bilayer. Recent molecular dynamics (MD) simulations showed that key serine residues in the hydrophobic helix interact directly with TAG within the membrane. Given its position at the center of the Seipin ring, this hydrophobic helix effectively concentrates TAG molecules, thereby facilitating lens formation and LD budding[29,30]. This hydrophobic helix was also shown to bind to Promethin/LDAF1[30,31], a conserved ER membrane protein homologous to yeast Ldo45[32–34].

In contrast to mammalian and fly Seipins, the function of yeast Seipin Sei1 requires the ER membrane protein Ldb16, and LD defects observed in sei1Δ, ldb16Δ and sei1Δldb16Δ cells are indistinguishable[18,35]. Curiously, expression of human Seipin in yeast restores normal LD formation in sei1Δ ldb16Δ double mutants[18], indicating that in yeast, Seipin function is distributed between two polypeptides.

Here, we explored this unique organization of yeast Seipin to define the mechanism of Seipin-mediated LD formation. We show that Sei1 forms a homodecameric ring that scaffolds and positions Ldb16, with the latter emerging as the primary TAG binder via specific hydroxyl-containing residues, as proposed for

human Seipin. Our structure revealed a structural element, the locking helix, which enables a distinctive arrangement of Sei1 TM segments, critical for both Ldb16 stability and the initial recruitment of TAG to the Sei1/Ldb16 complex. We propose that the dual TAG interaction through the TM segments and central ring elements define a unifying mechanism for Seipin-mediated LD formation and lipid storage.

## Results

**Cryo-electron microscopy structure of yeast Sei1.** To gain insight into the mechanism of LD assembly by Seipin we determined the structure of the yeast Seipin Sei1. A functional C-terminally 3xFLAG tagged Sei1 (Sei1-FLAG) was overexpressed using a galactose-inducible promoter in S. cerevisiae. Sei1-FLAG was affinity purified from crude membranes solubilized in dodecyl maltoside (DDM), and supplemented with cholesterol hemisuccinate (CHS), followed by size-exclusion chromatography. Sei1-FLAG eluted in a single high molecular weight peak (Fig. S1A and B), which exhibited large homogeneous particles when analyzed by negative stain EM (Fig. S1C) suggesting it to be an oligomer. Next, the peak fraction of Sei1-FLAG was analyzed by single particle cryo-EM (Fig. 1A and S1D–F). The structure of Sei1 was determined to a resolution of 2.7 Å (Fig. 1A, B and S1D-F). The Seipin ring is assembled from 10 Sei1 protomers with a diameter of 140 Å in the outer and 25 Å in the inner rings (Fig. 1A, B). The Sei1 decamer contrasted with the undecameric and dodecameric rings assembled by human[27] and fly[28] Seipins, respectively. Sei1 protomers are characterized by an 8-strand β-sandwich capped by two orthogonal short helices (α1 and α2) at the inner surface of the ring protruding slightly towards the membrane side (Fig. 1C). The Sei1 β-sandwich fold (Fig. 1C) is reminiscent of the one present in the cholesterol binding NPC2[36] (Fig. S1G), as observed for the β-sandwiches in human[27] and fly[28] Seipin luminal domains (Fig. 1D). Despite the differences in subunit number, a ring-like arrangement composed of β-sandwich motifs appears to be a general feature for Seipin function across eukaryotes.

Besides the luminal domain and in contrast to previous studies, our Sei1 structure also revealed regions proximal to and within the membrane. On the outer diameter of the Sei1 ring, just underneath the β-sandwich, we observed a short helix parallel to the plane of the membrane (α3) that sits on top of the two TM segments (Fig. 1C). Although the two TM segments are encoded by amino acids at the very N- and C-terminal of Sei1 sequence, they come into close proximity and show extensive interactions in the folded polypeptide (Fig. 1C).

**The Sei1 luminal domain lacks a hydrophobic helix and the ability to concentrate TAG.** A luminal helical region on the inner surface of the Seipin ring (α1, α2), projecting towards the ER membrane (Figs. 1D and 2A), is present across eukaryotes. However, we observed marked differences in the size, position, and properties of the Sei1 helical region when compared to the equivalent portion of human and fly Seipins. In human and fly Seipins the helix is large, slightly kinked (Fig. 2A), and combines hydrophobic (Fig. 2A) and uncharged (Fig. S2A) amino acids, hence it has been referred to as hydrophobic helix[27,28]. Due to its size and amino acid composition, the hydrophobic helix of human and fly Seipin insert in the luminal leaflet of the bilayer (Fig. 2B). Importantly, recent MD simulations indicate a functional role of the hydrophobic helix within the membrane through conserved serine residues that interact directly with TAG carbonyl groups, resulting in effective concentration of TAG within the Seipin rings[29,30]. In contrast, the helical region in yeast

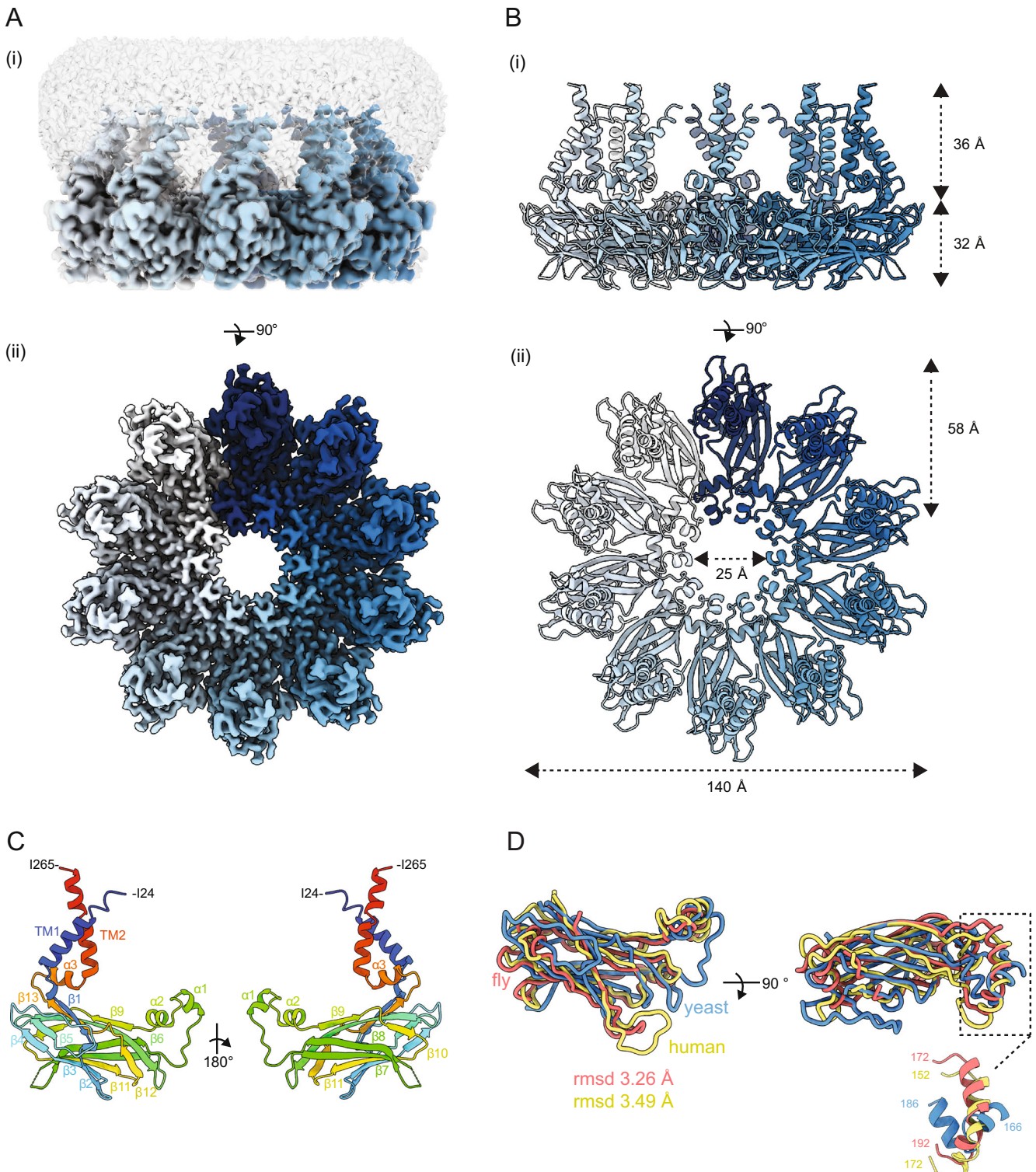

**Fig. 1 Structure of the yeast Seipin Sei1. A** Cryo-EM map of Sei1 homodecamer shown as (i) side view, depicting protein density (contour level of 0.007) with each Sei1 protomer colored in shades of blue, and the surrounding detergent micelle shown in transparent gray (contour level of 0.005) or (ii) 90-degree rotated view looking down from the cytosol, with detergent micelle omitted for clarity (contour level of 0.007). **B** Cartoon representation of the Sei1 homodecamer model depicted as (i) side view or (ii) 90-degree rotated view looking down from the cytosol, as depicted in (**A**). Individual Sei1 protomers are displayed in different shades of blue. **C** Cartoon of Sei1 protomer model colored as rainbow from N- (blue) to C-(red) termini. The various Sei1 domains are indicated: partial transmembrane helices (TM1 and TM2) capped by the short α3 helix; the luminal β-sandwich (β-strands 1–13); and central helices α1-2. **D** Structural alignment of the luminal domains of yeast (blue; residues 49–232), fly (residues 88-240 of PDB 6MLU; orange), and human (residues 60–219 of PDB 6DS5; yellow) Seipin protomers. Indicated RMSD is against the yeast protomer. Inset—detail of the luminal helices protruding towards the ER membrane.

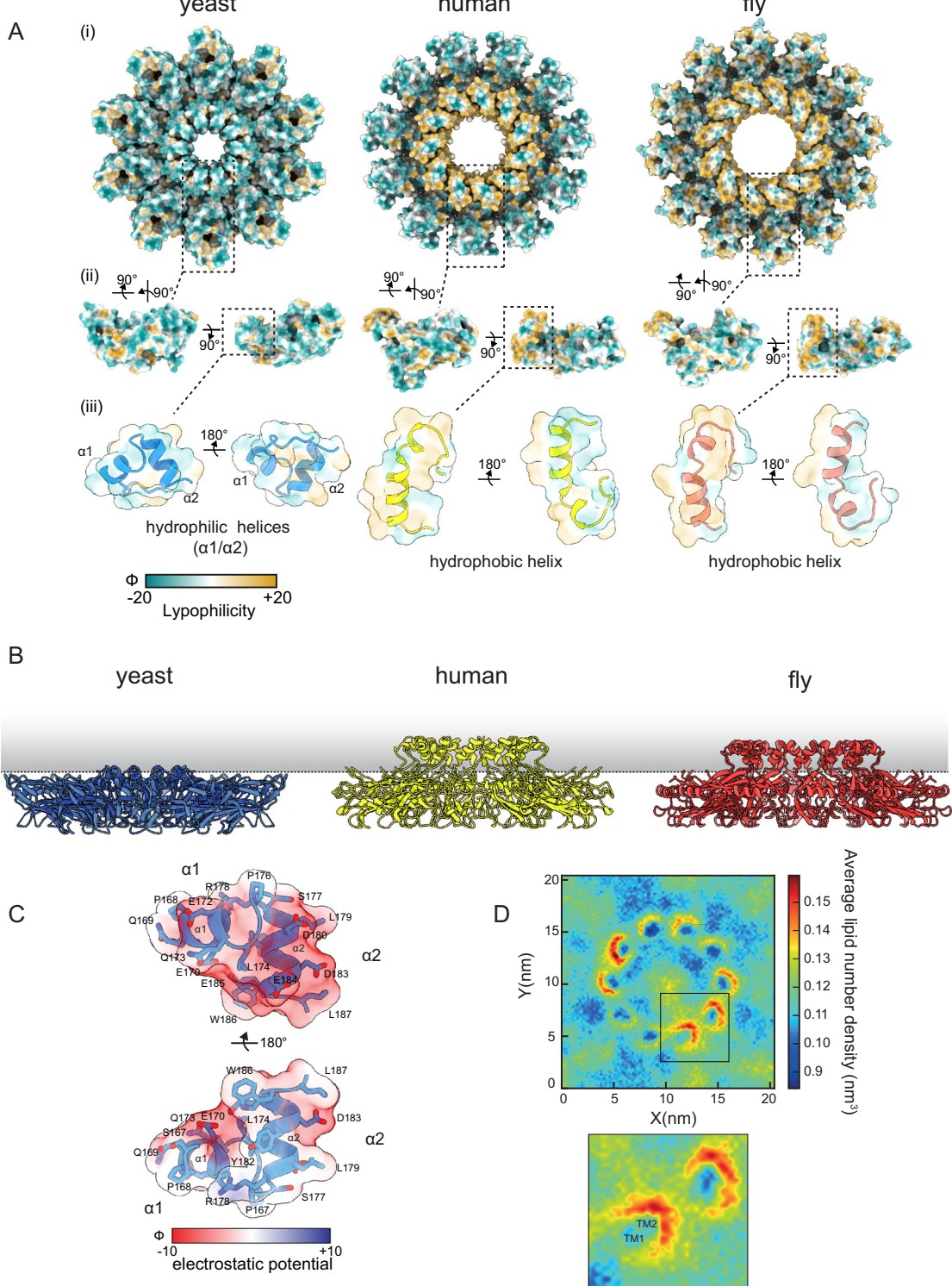

**Fig. 2 The Sei1 luminal domain lacks a hydrophobic central helix and the ability to concentrate TAG. A** Lipophilicity potential mapped to surface representations of yeast (left), human (middle), and fly (right) as viewed as (i) homodecamer assembly from the cytosol, (ii) individual protomers, or (iii) transparent overlay over zoomed in cartoon representation of the central α1-α2 helices. Surfaces are colored from hydrophilic (dark cyan) to hydrophobic (gold). Lipophilicity potential was generated from the mlp command of ChimeraX. **B** Side view of the luminal domains of yeast, human (PDB 6DS5) and fly (PDB 6MLU) Seipin in relation to the plane of the ER membrane (indicated by a dotted line). **C** Charge distribution of the yeast Sei1 central helices (α1, α2), depicted as a transparent Coulombic electrostatic potential surface representation (Red, negative charge; blue, positive charge; white, no charge) overlayed over a cartoon representation (light blue) to show acidic side chains. Electrostatic potential was generated from the coulombic command of ChimeraX. **D** Top View of coarse-grained MD simulations of Sei1 in a POPC membrane with 3% trioleylglycerol. Images depict average lipid number density of trioleylglycerol. Inset – zoom in of corresponding box showing positions of TM1 and TM2.

Sei1 luminal domain is split into two orthogonal small helices (α1 and α2) (Fig. 2A), and does not extend into the membrane (Fig. 2B). Furthermore, the Sei1 α1/ α2 region is enriched in charged residues (Fig. 2C, Fig. S2A). Coarse-grained MD simulations showed that TAG fails to concentrate within the rings of yeast Sei1, consistent with it lacking a membrane protruding hydrophobic helix (Fig. 2D). Curiously, mild but consistent TAG enrichment was observed in proximity of Sei1 TM segments suggesting a contribution of the membrane region in LD formation (Fig. 2D). In contrast, similar simulations using the human Seipin luminal domain showed dramatic TAG accumulation within the ring (Fig. S2B), as previously described[29,30]. Thus, the lack of membrane embedded hydrophobic helix indicates that yeast Sei1 uses an alternative mechanism for concentrating TAG during LD formation.

**Ldb16 complements Sei1 structure for LD formation**. In contrast to human and fly Seipin, Sei1-mediated LD formation depends on Ldb16, a yeast specific binding partner. This raises the possibility that Ldb16 complements the Sei1 structure by providing the hydrophobic helix required for TAG concentration within the ring. Consistent with this possibility, expression of human Seipin restores normal LD formation in yeast mutants lacking both Sei1 and Ldb16[18]. We attempted to explore this hypothesis using a structural approach on purified Sei1/Ldb16 complex. A functional fusion of Ldb16 to streptavidin binding protein (Ldb16-SBP) and Sei1-FLAG were co-overexpressed, and the complex was purified as described above. Although Sei1-FLAG and Ldb16-SBP co-purified (Fig. S3A, B), the EM map obtained with Sei1/Ldb16 complex was indistinguishable from the one obtained with Sei1 alone suggesting that Ldb16 was lost during sample vitrification.

To determine the positioning of Ldb16 in relation to the Sei1 ring we instead employed in vivo site-specific photo-crosslinking. A phenylalanine derivative carrying a photoreactive benzophenone (Bpa) was incorporated in Sei1-FLAG at positions specified by an amber stop codon, as described[37]. The photoreactive probe was individually placed at several positions within the first and second TM segments (TM1 and TM2, respectively) as well as in the luminal domain, including the two short hydrophilic α1/α2 helices unique to Sei1. Cells expressing Sei1-FLAG with individual Bpa probes and endogenous Ldb16 tagged with HA (Ldb16-HA) were UV-irradiated to trigger protein crosslinking. Strong ladder-like Sei1-Sei1 crosslinks were observed for probes inserted in the luminal region but not when Bpa was within TMs (Fig. 3A). This is in agreement with our structural data showing that interactions between Sei1 protomers occurs through the luminal domain. Prominent Sei1-Ldb16 crosslinks were also detected for specific Bpa probes in TM1, TM2 and central luminal region confirming an intimate relationship between the two proteins (Fig. 3A, B). Remarkably, the Sei1 α1/α2 helices showed strong crosslinks to Ldb16, particularly in residues pointing towards the center of the Sei1 ring. This indicates that Ldb16 is well-positioned to concentrate TAG. To test the importance of Sei1 α1/α2 helices in controlling Ldb16 positioning and LD formation we replaced this Sei1 region by a flexible linker (Sei1GGSGGS) (Fig. S3C). This mutant was expressed to levels comparable to WT Sei1 (Fig. S3D) and oligomerized efficiently (Fig. S3E). Ldb16 levels were not affected (Fig. S3D) and it co-precipitated with the mutant Sei1GGSGGS similarly as with WT (Fig. S3F), suggesting the formation of a stable complex. However, cells expressing Sei1GGSGGS had mildly enlarged LDs (Fig. 3C, D). Using a well-established assay to monitor de novo LD formation upon induction of a single TAG biosynthetic

gene[10], we observed that cells expressing Sei1GGSGGS were also defective in LD biogenesis, with a reduced number of LDs forming over time (Fig. S3G, H). Thus, Sei1 α1/α2 helices interact with Ldb16 possibly facilitating its positioning for TAG concentration during LD formation.

Ldb16 has N- and C-termini in the cytosol[18], and is predicted to have two TMs. In silico structure prediction by trRosetta[38] revealed an Ldb16 element with remarkable similarities to the TAG concentrating helices in human and fly Seipins (Fig. 3E). In between the two Ldb16 TM segments, there is a short helical region rich in serine (S53, S55, S62) and threonine (T52, T61, T63) residues, which through their hydroxyl groups could potentially be involved in TAG binding, as observed for human and fly Seipins.

To test the functional relevance of these hydroxyl-rich motifs, we generated several mutants in the S/T-residues. These mutants were expressed to normal levels (Fig. S3I) and bound endogenous Sei1 (Fig. S3J). Remarkably, these Ldb16 mutants displayed aberrant LDs (Fig. 3F, G) consistent with a role of the hydroxyl-containing residues in TAG concentration. Altogether, these data strongly support a model in which Sei1 ring extensively interacts with Ldb16 facilitating its positioning for TAG concentration through a mechanism analogous to human and fly Seipins.

**The Sei1 Locking Helix positions transmembrane segments and facilitates Ldb16 binding**. Ldb16 protein levels and stability depend on Sei1[18]. While our photo-crosslinking approach shows extensive interaction between the two proteins, deletion of Sei1 α1/α2 helices did not affect Ldb16 levels (Fig. S3D). This observation suggested that interactions with the TM regions are important for Ldb16 stability. To test this possibility various residues within Sei1 TM1 and TM2 were mutated. When expressed from the endogenous Sei1 promoter all these mutants displayed low steady state levels (Fig. S4A). Consequently, Ldb16 levels were also reduced (Fig. S4A) and LDs were aberrant (Fig. S4B). However, if expressed from the alcohol dehydrogenase promoter (ADH1p), a strong constitutive promoter, all Sei1 TM mutants showed levels comparable to WT Sei1 (Fig. S4C). Importantly, for most mutants this also resulted in the restoration of Ldb16 levels (Fig. S4C) and normal LDs (Fig. S4B).

The notable exception was the mutation of tyrosine residues at positions 37 and 41 (Y37 and Y41, respectively). Proximal to the luminal face of the membrane, Y37/Y41 establish methionine-aromatic interactions with a methionine residue (M240) in a short helix (α3) in between the luminal β-sandwich and the TM region, hereafter called locking helix (LH) (Fig. 4A). In cells expressing Sei1LL, a mutant where the aromatic residues (Y37/Y41) were replaced by leucines (L), Ldb16 levels remained low (Fig. S4C). Similarly, cells expressing Sei1ΔLH, where a flexible linker replaced the LH (Fig. S4D), also failed to stabilize Ldb16 (Fig. S4C). Moreover, Sei1ΔLH and Sei1LL cells displayed aberrant LDs however, in comparison to sei1Δ mutant, the frequency of supersized LDs was reduced (Fig. 4B, C). Thus, disruption of the methionine-aromatic interactions by mutating the LH or key aromatic residues in TM1 results in LD morphology defects.

To gain further insight of how methionine-aromatic interactions influenced the behavior of Sei1 TM segments we employed MD. Atomistic simulations reveal that in WT Sei1, TM1 and TM2 adopt a stable conformation and maintain a constant angle (Fig. 4D) indicating that they move largely as a single unit (Fig. 4E). This coupling of Sei1 TM movement is impaired upon

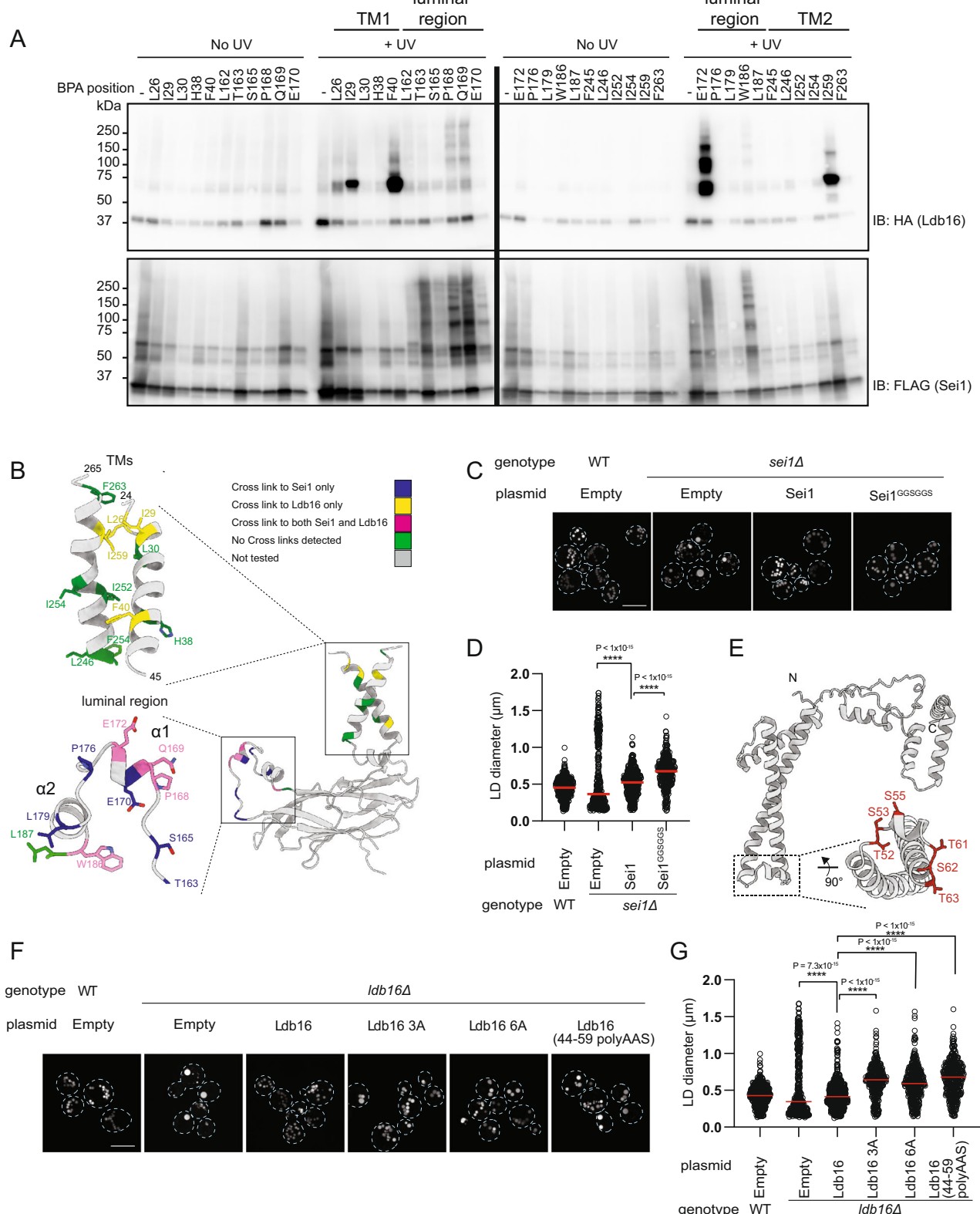

disruption of the methionine-aromatic interactions, in Sei1$^{\Delta LH}$ and Sei1$^{LL}$ (Fig. 4E). Besides their uncoordinated movement, TM segments in Sei1$^{\Delta LH}$ and Sei1$^{LL}$ appear to explore a higher membrane area. Importantly, the relative position of the TM domains to the Sei1 luminal domain did not appear to be affected by the mutations (Fig. S4E). To further demonstrate that the effect of methionine-aromatic interactions was restricted to the positioning and dynamics of the TM segments we solved the structure of Sei1$^{\Delta LH}$. FLAG-tagged Sei1$^{\Delta LH}$ was expressed and purified as above (Fig. S4F and G), and analyzed

**Fig. 3 Ldb16 complements Sei1 structure for LD formation. A** *sei1Δ* cells expressing endogenously HA-tagged Ldb16 and plasmid-borne *SEI1-FLAG* with a photoreactive Bpa at the indicated positions were subjected to UV irradiation. Non-irradiated cells were used as controls. Solubilized membranes were subjected to immunoprecipitation with anti-FLAG antibodies, and bound proteins were analyzed by immunoblotting with FLAG and HA antibodies. **B** Schematic representation of the Sei1-Sei1 and Sei1-Ldb16 site-specific photo-crosslinks obtained in (**A**). **C** Analysis of LDs in cells with the indicated genotype after staining with the neutral lipid dye BODIPY 493/503. Scale bar corresponds to 5 µm. **D** Quantification of LD diameter of cells shown in (**C**). At least 100 LDs were analysed for a minimum of 3 biological repeats. Red bars represent median diameter. $n = 3$. Difference in distribution of LD size was tested using a two sided Kolmogorov-Smirnov test (**** $p < 0.0001$, *n.s.* non-significant). **E** Cartoon of Ldb16 In silico structural prediction by trRosetta. Inset shows a short helical element rich in hydroxylated residues (in red). **F** Analysis of LDs in cells with the indicated genotype after staining with the neutral lipid dye BODIPY493/503. Scale bar corresponds to 5 µm. **G** Quantification of LD diameter of cells shown in (**F**). At least 100 LDs were analysed for a minimum of 3 biological repeats. Red bars represent median diameter. $n = 3$. Difference in distribution of LD size was tested using a two sided Kolmogorov-Smirnov test (**** $p < 0.0001$, *n.s.* non-significant).

by cryo-EM (Fig. 4F and Fig. S4H–J). As predicted by MD, the luminal domains of Sei1 and Sei1$^{ΔLH}$ were indistinguishable (Fig. 4G). However, in Sei1$^{ΔLH}$ the TM domains could not be resolved (Fig. 4F, G), likely due their increased movement. Together, these data show that the LH has a central role in controlling the positioning and dynamics of Sei1 TM segments, both important for Ldb16 stability. Moreover, our data indicate that Sei1 luminal and TM domains independently contribute to LD formation.

**Sei1 transmembrane segments contribute to LD formation.** We identified a critical role of Sei1 TMs in LD formation through the stabilization of Ldb16. Our coarse-grained MD simulations also showed that Sei1 TMs, unlike its luminal domain, are able to interact with TAG (Fig. 2D). Thus, we wondered if Sei1 TMs, through this potential TAG binding activity, contributed to LD formation independently of Ldb16. To test this possibility, we asked whether Sei1 could modify LD morphology in the absence of Ldb16. Aberrant LDs observed in *sei1Δ*, *ldb16Δ* and *sei1Δldb16Δ* mutant cells are largely indistinguishable[18,19,35]. Similarly, aberrant LD morphology of *sei1Δldb16Δ* was unmodified by plasmid-borne Sei1 expression from its endogenous promoter (Fig. S5A). However, Sei1 expression from the strong *ADH1*p resulted in striking changes in LD morphology in *sei1Δldb16Δ* cells (Fig. 5A, B and S5A). While in this condition LDs were still aberrant, they appeared smaller and clustered at the expense of supersized LDs, which were present at much lower frequency (Fig. 5A, B). This result shows that Sei1 can modify TAG partition into LDs independently of Ldb16. To test whether this activity could be ascribed to Sei1 TMs, we generated Sei1-$^{SecTM}$ and Sei1$^{WALP}$ mutants, in which Sei1 TM1 and TM2 were replaced respectively, by Sec61 TM1 and a synthetic WALP TM[39], composed of alternating leucines and alanines, and flanked by tryptophan residues (Fig. S5B). Strikingly, Sei1$^{SecTM}$ and Sei1$^{WALP}$ failed to modify LD morphology in *sei1Δldb16Δ* (Fig. 5A, B) as well as in *sei1Δ* cells (Fig. S5C, D). Both Sei1$^{SecTM}$ and Sei1$^{WALP}$ were expressed to the expected levels (Fig. S5E) and formed oligomers (Fig. S5F), consistent with the involvement of the luminal domain in Sei1 ring assembly. This result confirms that Sei1 luminal domain does not interact with TAG. Moreover, it indicates that Sei1 TMs contribute to the partition of TAG into LDs.

The swap of TMs in Sei1$^{SecTM}$ and Sei1$^{WALP}$ likely affects the methionine-aromatic interactions with the LH. Thus, besides the amino acid sequence, Sei1$^{SecTM}$ and Sei1$^{WALP}$ might also differ in the dynamics of the TMs. To identify the critical TM determinants for the TAG interactions we tested Sei1$^{ΔLH}$, a mutant where only TM dynamics is affected. Like WT Sei1, expression of Sei1$^{ΔLH}$ triggered dramatic changes in LD morphology in *sei1Δldb16Δ* cells (Fig. 5A, B). Thus, amino acid

composition but not dynamics appears to be critical for the interactions of Sei1 TM with TAG. Finally, we tested whether the ability of Sei1 TMs to interact with TAG is conserved in human Seipin. Remarkably, expression of Sei1$^{hsTM}$, a chimeric protein between Sei1 luminal domain and the TMs of human Seipin (Fig. S5G), triggered LD morphological changes in *sei1Δldb16Δ* cells as observed by expression of WT Sei1 (Fig. 5A, B). As expected, expression of WT human Seipin (WT hs) in *sei1Δldb16Δ* cells restored normal LDs (Fig. 5A, B)[18]. Thus, interactions between TAG and the TMs of Sei1, and perhaps other Seipins, contribute to LD formation and morphology.

**Discussion**

Seipins have conserved roles in LD biogenesis but the mechanistic basis for their function is poorly understood. Here, we identify the determinants within the Sei1/Ldb16 Seipin complex essential for proper LD formation. We show that regions both in the membrane and lumen of the ER play critical roles, likely through their ability to interact with TAG (Fig. 6). Besides binding and stabilizing Ldb16, the Sei1 TMs increase the local concentration of TAG in the proximity of the Seipin complex (Fig. 6, Step 1), possibly controlling its access to the homodecameric Sei1 ring (Fig. 6, Step 2). Once inside the ring, hydroxyl-residues in a short Ldb16 helix interact with TAG (Fig. 6, Step 3), increasing its local concentration and promoting phase separation into a lens like structure. As discussed below, we propose that these sequential TAG interactions are general steps in Seipin-mediated neutral lipid recruitment required for LD formation and lipid homeostasis.

The Sei1 luminal domain forms a homooligomeric ring, like other Seipins[27,28]. In human Seipin, the ring facilitates TAG concentration via interactions with two highly conserved hydroxyl-containing residues in a central hydrophobic helix[28–30]. In contrast, we showed that the Sei1 luminal ring lacks a hydrophobic helix and does not concentrate TAG. Instead, we found that Ldb16 complements the Sei1 luminal domain through a short helical region rich in hydroxyl-residues, thus providing the TAG interacting moieties to the yeast Seipin complex. The ring assembly of Sei1 luminal domain facilitates this interaction by acting as a scaffold for Ldb16. The ER membrane protein Ldo45 was shown to interact with Sei1/Ldb16 complex and modulate its activity[33,34]. Moreover, the human Ldo45 homolog, LDAF1/Promethin, interacts with the hydrophobic helix of Seipin luminal domain, an interaction that is modulated by TAG[30–32]. Thus, it is possible that Ldo45 regulates the Sei1/Ldb16 complex by controlling Ldb16 position and/or TAG binding activity.

We identified critical roles for the Sei1 TMs during LD formation, by binding independently Ldb16 and TAG. The binding to Ldb16 depends on the unique TM1/TM2 arrangement maintained via methionine-aromatic interactions between the LH and

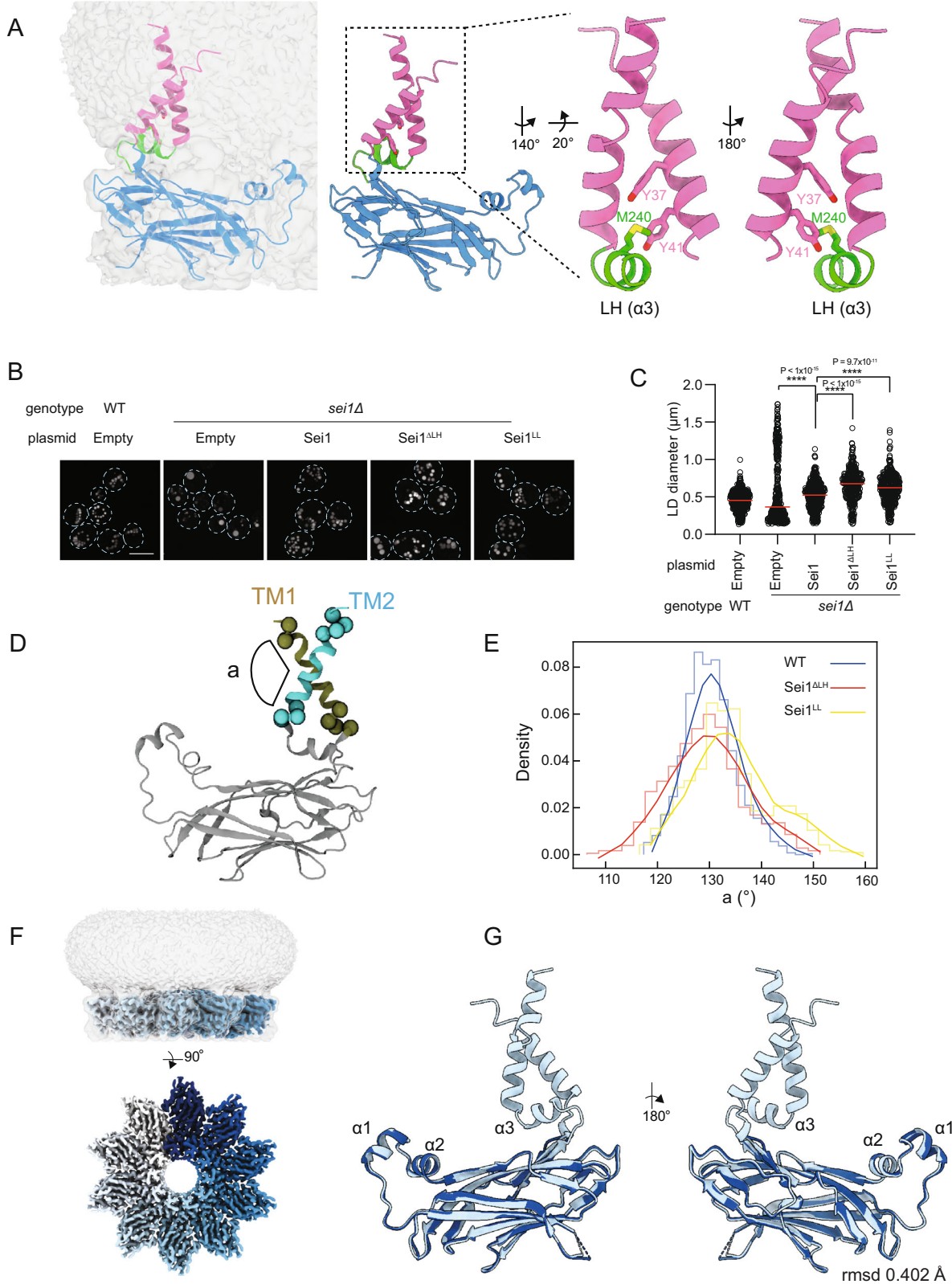

TM1, respectively. While the TMs of human and fly Seipins were not resolved in previous structural studies[27,28], in silico analysis predicts the existence of a short helical domain, equivalent to the LH, in those Seipins. It will be interesting to test the functional

relevance of this helical domain, and whether it regulates the relative arrangement of the TMs in other Seipins.

In addition, Sei1 TMs also modulate the partitioning of TAG into LDs. By promoting only mild TAG concentration, Sei1 TMs

**Fig. 4 Sei1 Locking Helix positions transmembrane segments and facilitates Ldb16 binding. A** Sei1 protomer model shown (left) in the context of the map at low contour (level of 0.005) within detergent micelle or (middle) without map overlay. Inset (right) depicts the locking helix (LH, green) and TM1 and TM2 (pink). The LH M240 (yellow) establishes methionine-aromatic interaction with Y37 and Y41 (red) in TM1. **B** Analysis of LDs in cells with the indicated genotype after staining with the neutral lipid dye BODIPY 493/503. Scale bar corresponds to 5 μm. **C** Quantification of LD diameter of cells shown in (**B**). At least 100 LDs were analysed for a minimum of 3 biological repeats. Red bars represent median diameter. $n = 3$. Difference in distribution of LD size was tested using a two sided Kolmogorov-Smirnov test (****$p < 0.0001$, *n.s.* non-significant). **D** Schematic representation of the angle (**a**) between Sei1 TM1 and TM2 analyzed by atomistic MD simulations in (**E**). **E** Computed angles between TMs 1 and 2 during 3 × 120 ns atomistic MD simulations of Sei1, Sei1$^{\Delta LH}$ and Sei1$^{LL}$ with the first 20 ns discarded as equilibration. The data has been binned into 1D histograms (thin line) with a line fitted through the bin centers and smoothed using a Savitzky-Golay filter with a step of 5 (thick line). **F** Cryo-EM map of Sei1$^{\Delta LH}$ homodecamer shown as side view (top), depicting luminal domain protein density (contour level of 0.05) with each Sei1 protomer colored in shades of blue, and the surrounding detergent micelle shown in transparent gray (contour level of 0.005) or 90-degree rotated view looking down from the cytosol (bottom), with detergent micelle omitted for clarity. **G** Structural alignment of the luminal domains of Sei1 (light blue) and Sei1$^{\Delta LH}$ (dark blue).

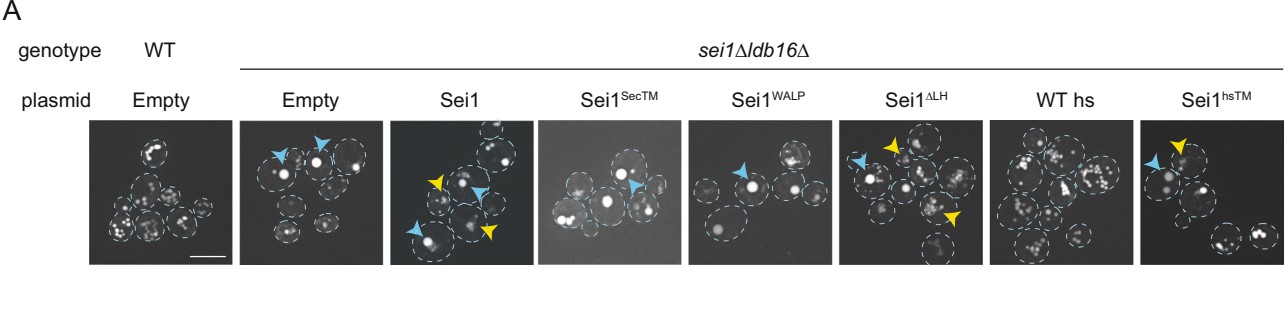

A

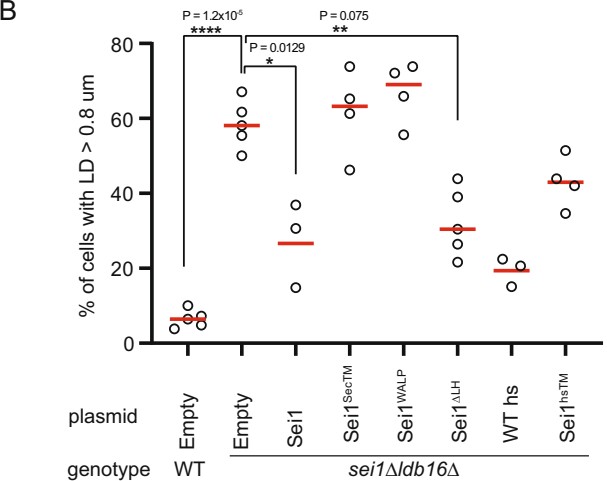

**Fig. 5 Sei1 transmembrane segments contribute to LD formation. A** Analysis of LDs in cells with the indicated genotype after staining with the neutral lipid dye Bodipy493/503. Scale bar corresponds to 5 μm. Blue and yellow arrowheads indicate supersized LDs (ssLDs) and LD aggregates, respectively. **B** Quantification of ssLDs in cells with the indicated genotype. At least 50 cells were analysed per strain for a minimum of 3 biological replicates (WT empty vector, $n = 5$; sei1Δldb16Δ empty vector and Sei1$^{\Delta LH}$, $n = 5$; sei1Δldb16Δ Sei1$^{SecTM}$ Sei1$^{WALP}$ and Sei1$^{hsTM}$, $n = 4$; sei1Δldb16Δ Sei1 and WThs, $n = 3$). Red bars represent median percentage. Difference in frequency of ssLDs was tested using a two sided paired t-test (*$p < 0.05$, **$p < 0.01$, ****$p < 0.0001$, *n.s.* non-significant).

are incapable to promote lens formation. Instead, they may regulate the kinetics of TAG entry into the Sei1 ring, as proposed for human Seipin[29]. Such an activity may also be important to coordinate lens growth with incorporation of monolayer phospholipids during LD expansion. Seipin rings not in contact with lenses are highly mobile in the ER[17,20,40]. Thus, by encountering TMs of dynamic Seipin rings, TAG molecules dispersed in the membrane may be concentrated at specific ER regions for proper LD assembly, a function consistent with the role of Seipin in averting lipotoxicity. Besides TAG, Sei1 TMs may also affect the distribution of other lipids in the ER bilayer. Consistent with this possibility, in silico predictions indicate that the TMs of human Seipin promote localized enrichment of diacylglycerol (DAG) and phosphatidylethanolamine (PE)[29]. Given the impact of ER bilayer composition in LD biogenesis[16,41–44], Seipins TMs can contribute for a local membrane environment conductive to proper LD formation. Our findings provide a mechanistic framework for Seipin function, which will guide future structural and biophysical studies.

## Methods

**Reagents**. BODIPY493/503 was purchased from Invitrogen. Antibodies used in this study were anti -FLAG M2-Peroxidase (HRP), Clone M2-A8592 (Sigma Aldrich) product number A8592, anti-HA High affinity (clone 3F10) product number 11867431001 (Roche), PGK1 Monoclonal (22C5D8) product number 459250 (Invitrogen), DPM1 monoclonal (5C5A7) product number A6429

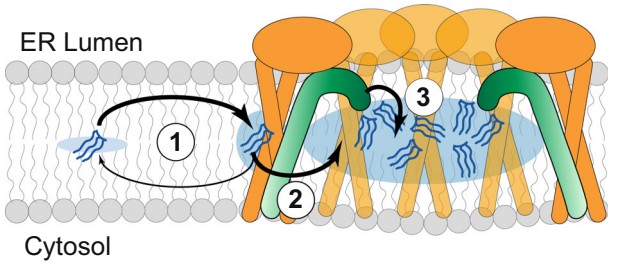

**Fig. 6 A model for LD formation by the Sei1-Ldb16 complex.** Sequential TAG interactions mediate LD assembly by the Sei1-Ldb16 complex. In the ER bilayer, TAG molecules (blue) concentrate in proximity of Seipin oligomers (orange) via weak interaction with Sei1 TMs. TAG molecules within the ring interact strongly with Ldb16 (green) hydroxyl-containing residues, facilitating TAG coalescence and lens formation (see text for details).

(Thermo Fisher Scientific)). Polyclonal anti-Sei1 (rabbit), anti-Ldb16 (rabbit) antibodies were previously described[33].

**Yeast strains and plasmids.** The strains used are isogenic either to BY4741 (*MATa ura3Δ0 his3Δ1 leu2Δ0 met15Δ0*) or FY251 (*MATa ura3-52 his3Δ200 leu2Δ1 trp1Δ63*) and are listed in the Table S1. Tagging of proteins and individual gene deletions were performed by standard PCR-based homologous recombination[45] and standard yeast molecular genetics protocols[46].

Plasmids used are based on pRS316, pRS416 pRS423 or pRS426[47,48] and listed in Table S2. 3xFLAG-tagged Sei1 was expressed from the native Sei1 promotor (494 bp upstream of the *SEI1* ORF) or an *ADH1* promotor, and followed by the *ADH1* terminator. 3xFLAG-tagged Ldb16 was expressed from the native promotor (459 bp upstream of *LDB16* ORF) followed by the *ADH1*-terminator. Primers used in this study are listed in Table S3.

**Culture conditions.** Cells were cultured in synthetic defined glucose media (SD), unless otherwise indicated. SD contained per liter: 6.7 g yeast nitrogen base with ammonium sulfate (YNB; MP biomedicals), 0.6 g complete supplement mixture without histidine, leucine, tryptophan, and uracil (CSM-HIS-LEU-TRP-URA; MP biomedicals), and 20 g glucose (Sigma-Aldrich). Media was supplemented with histidine (60 μM), leucine (1.68 mM), uracil (0.2 mM) and tryptophan (0.4 mM) as required. For inositol free media, YNB devoid of inositol (MP biomedical) was used.

All cultures were incubated at 30 °C with shaking at 200 rpm. Culture density was determined by measuring turbidity at 600 nm (OD$_{600}$) using a GENESYS 10 S UV-VIS spectrophotometer (Thermo Scientific).

**Protein expression and purification for structural analysis.** For protein over-expression, *sei1Δldb16Δ* mutants were transformed with a plasmid encoding Sei1-FLAG, Sei1$^{ΔLH}$-FLAG or both Sei1-FLAG and Ldb16-SBP[49]. Cells were grown and protein was expressed as described[50]. Cell pellets (~150 g) were harvested by centrifugation, washed with water and lysis buffer (50 mM Tris.HCl [pH7.4], 200 mM NaCl, 1 mM EDTA). Cells were resuspended in 100 mL of lysis buffer with 1 mM phenylmethylsulfonylfluoride (PMSF) (Roche) and 1.5 μM of Pepstatin A (Sigma-Aldrich) and transferred to a bead beater chamber (BioSpec) containing ~150 g glass beads (0.5 mm diameter; BioSpec). Bead beater chamber was assembled with an ice water jacket. Lysis was induced by 40 cycles of 30 s on/off. Glass beads were removed by filtration and lysates cleared by low-speed spinning at 2000 g for 30 min. Total membrane fraction was prepared by centrifugation (185511.4 × g in a Ti-45 for 45 min). and washed with lysis buffer. The membrane pellet was solubilized for 4 h in 195 mL of lysis buffer supplemented with 1%(w/v) of DDM (Anatrace) and 0.1% (w/v) CHS (Anatrace), 1 mM PMSF (Roche), 1.5 μM Pepstatin A (Sigma-Aldrich). Non-solubilized material was removed by centrifugation (185511.4 × g in a Ti-45 for 30 min). 4 mL of FLAG matrix -M2 affinity gel - A2220 (Sigma Aldrich) was added to the solubilized membranes and incubated at 4 °C overnight. After incubation, the material was transferred to 20 mL gravity columns and beads were washed with 10 column volumes of Akta buffer (50 mM Tris.HCl [pH7.4], 200 mM NaCl, 1 mM EDTA, 0.015% DDM, 0.0015% CHS) by gravity flow. Bound proteins were eluted with Akta buffer in 5 × 3 mL fractions containing 0.2 μg/mL 3xFLAG-peptide for the first two fractions and 0.4 μg/mL 3xFLAG-peptide for the last three rounds. Eluted material was concentrated using 100 kDa cut off centrifugal filters (Amicon Ultra, Merck) until the volume reached below 2 mL. The concentrated material was run with an AKTA

Pure (SEC) (GE Healthcare) over a 24 mL Superose 6 10/300 GL size exclusion column in Akta buffer, at 0.5 mL/min, collecting 1 mL aliquots

**Negative stain.** For negative stain EM analysis, 8 μL of purified Sei1 (protein concentration ~20 ng/μL) was added to glow discharged 300 mesh carbon support films (TAAB) and immersed in 20 μL water, twice in 20 μL of 2% uranyl acetate, and dried for at least 5 min before use. Grids were imaged in a FEI Tecnai T12 transmission electron microscope.

**Cryo-EM sample preparation and data acquisition.** Four microliters of purified Sei1 or Sei1-Ldb16 complex at a concentration of 4 mg/mL was adsorbed to glow-discharged gold UltrAuFoil grids (300 mesh, R1.2/1.3) for 10 s. Grids were then blotted for 2 s at 100% humidity at 9 °C and frozen in liquid ethane using a Vitrobot Mark IV (Thermo Fisher Scientific). Data were collected in counting mode on a Titan Krios G3 (FEI) operating at 300 kV with a GIF energy filter (Gatan) and K2 Summit detector (Gatan) using a pixel size of 0.822 Å, a dose rate of 6 e$^-$ per Å$^2$ per s and an exposure of 8 s, corresponding to a total dose of 48 e$^-$ per Å$^2$ collected over 32 fractions.

Four microliters of purified Sei1Δ231-243 (Sei1$^{ΔLH}$) at a concentration of 7.8 mg/ml was adsorbed to a glow-discharged gold UltrAuFoil grid (300 mesh, R1.2/1.3) for 10 s. Grids were then blotted for 2 s at 100% humidity at 6 °C and frozen in liquid ethane using a Vitrobot Mark IV (Thermo Fisher Scientific). Data were collected in counted super-resolution mode on a Titan Krios G3 (FEI) operating at 300 kV with a BioQuantum imaging filter (Gatan) and K3 direct detection camera (Gatan) at 105,000× magnification, physical pixel size of 0.832 Å. Data were collected at a dose rate of 22.2 e− per Å$^2$ per s and an exposure time of 2.66 s, corresponding to a total dose of 59.1 e− per Å2 collected over 40 fractions.

**Cryo-EM data processing.** Initial micrograph processing was performed in real time using the SIMPLE pipeline[51], using SIMPLE-unblur for patched motion correction, SIMPLE-CTFFIND for CTF estimation and SIMPLE-picker for particle picking. Resolution estimates were derived from gold-standard Fourier shell correlations (FSCs) using the 0.143 criterion as calculated within RELION-3.1[52]. Local resolution estimations were calculated within RELION-3.1.

1,369,344 combined particles from Sei1 (2,008 movies) and Sei1-Ldb16 (6,499 movies) datasets were extracted in 300 × 300 pixel boxes and subjected to initial 2D classification (SIMPLE_cleanup2D) to remove junk particles. A subset of 81,812 recovered particles were used to generate a C10-symmetrized ab initio model (SIMPLE_initial_3Dmodel). All further downstream processing was performed in RELION-3.1[52]. The initial model was lowpass filtered to 60 Å and used as reference for unmasked 3D classification (7.5° sampling, 15 iterations, 3 classes, C10 symmetry) against the same particle subset. The map corresponding to the class with highest particle distribution (73.9% of total particles) was lowpass filtered to 40 Å and used as reference for unmasked 3D classification (7.5° sampling, 15 iterations, 3 classes, C10 symmetry) using the cleaned particle subset. Particles (234,989) and map belonging to the dominant class (46.1% particle distribution) were subjected to 3D auto-refinement in C10 using a 15 Å lowpass filter for the reference map and a mask encompassing all protein and detergent density. This generated a volume with global resolution estimate of 3.3 Å. Per-particle defocus refinement and beamtilt estimation followed by Bayesian particle polishing in a larger box (432 × 432) further improved map quality to 2.7 Å. Sei1 and Sei1-Ldb16 datasets were initially processed independently until it became apparent that volumes generated from refinements for either datasets were identical, with no additional density corresponding to Ldb16. To boost the number of particles going into the final reconstructions and improve map quality, the two datasets were combined and processed together. Data processing workflow is presented in Fig. S1D.

In total 1,198,818 particles from the Sei1$^{ΔLH}$ dataset (7,077 movies) were extracted in 300 × 300 pixel boxes and subjected to initial 2D classification (SIMPLE_cleanup2D) to remove junk particles followed with further processing in RELION-3.1[52]. Unmasked 3D classification (7.5° sampling, 15 iterations, 4 classes, C1 symmetry) was performed using the cleanup2D-recovered particles (431,932), against a 40 Å lowpass filtered Sei1 reference map. 3D auto-refinement (in C1) with a mask encompassing all protein and detergent was performed independently on the two most populated classes, resulting in 4.0 Å and 4.5 Å maps, respectively. No significant differences in map density were observed across both refined volumes. Particles belonging to both classes were therefore combined (260,532 total particles), reextracted and recentred in 432 × 432 boxes, and subjected to masked 3D auto-refinement with C10 symmetry, resulting in a 3.3 Å map. Data processing workflow is presented in Fig. S4H.

**Model building and refinement.** The atomic model of Sei1 (residues 24–132, 145–265; Table S4) was built de novo from the 2.7 Å local-resolution filtered and sharpened map following several rounds of manual building using Coot v.0.944[53] and real-space refinement in PHENIX v.1.18.2-387445[54] using secondary structure, NCS, rotamer, and Ramachandran restraints.

The atomic model of Sei1$^{ΔLH}$ (residues 49–132, 145–230; Table S4) was generated by rigid-body fitting the Sei1 model into the 3.3 Å Sei1$^{ΔLH}$ local-resolution filtered and sharpened map followed by manual building in Coot and

multiple rounds of real-space refinement in both Coot and PHENIX. Both Sei1 and Sei1$^{\Delta LH}$ models were validated using MolProbity[55] within PHENIX. Figures were prepared using UCSF ChimeraX v.1.149, and PyMOL v.2.4.0. Structural alignments of yeast Sei1 and Sei1$^{\Delta LH}$ protomer models were performed within ChimeraX using the MatchMaker command. Structural alignments of yeast Sei1 protomers (residues 49-232) with protomers of fly (PDB 6MLU; residues 88-240) or human (PDB 6DS5; residues 60-219) Seipin were performed within CCP4[56] using SSM superposition within the superpose[57] program.

**Molecular dynamics simulations**. For the coarse-grained MD, the coordinates of yeast Sei1 were converted to the Martini 2.2 force field[58,59]. Alternatively, the human Seipin luminal domain (PDB 6DS5) was used, with the TM regions modeled as per the yeast Sei1 TMs, using Swiss-Model[60], with the final model available at osf.io/5depa. Harmonic bonds of 500 kJ mol$^{-1}$ nm$^{-2}$ were applied between all protein backbone beads within 1 nm. For the human Seipin simulations, additional flat-bottomed distance restraints of 1000 kJ mol$^{-1}$ nm$^{-2}$ were applied using PLUMED 2.4.4[61] between the backbone beads of Val-248 of one protomer, with Leu-39, Leu-167, and Val-248 of the next protomer.

Proteins were embedded into membranes composed of 3% TAG (trioleylglycerol) and 97% POPC, using the *insane* protocol[62]. All systems were solvated with Martini waters and Na$^{+}$ and Cl$^{-}$ ions to a neutral charge and a 0.15 M concentration. Systems were minimized using the steepest descents method, followed by 1 ns equilibration with 5 fs time steps, then by 100 ns equilibration with 20 fs time steps, before 5× ca. 13 μs (yeast) or 3 × 5 μs (human) production simulations using 20 fs time steps, all in the NPT ensemble with the V-rescale thermostat and semi-isotropic Parrinello-Rahman pressure coupling[63,64].

For the atomistic simulations, short CG sims were run of the yeast Sei1 complex in a POPC membrane, either using the WT system or with the mutations added manually. Snapshots were then converted to an atomistic description using the *cg2at* program[65], before production simulations of 3 × 120 ns per system using a 2 fs time step in the NPT ensemble with the V-rescale thermostat and semi-isotropic Parrinello-Rahman pressure coupling[63,64].

All simulations were run in Gromacs[66]. 2D densities were computed using gmx density from the Gromacs software, and plotted in Matplotlib[67]. Images were made in VMD[68].

**In vivo site-specific crosslinking**. Site specific crosslinking was conducted as previously described[69]. Briefly, sei1Δ cells were transformed with two plasmids, one encoding both for a modified tRNA synthetase capable of charging the unnatural amino acid benzoyl phenylalanine (BPA) on a tRNA as well as amber stop codon suppressor tRNA, and a second plasmid encoding ADH1-promotor expressed Sei1-FLAG with individual amber codons. Cells carrying both plasmids were pre-cultured in SD for 8 h, transferred to 100 mL of the same media supplemented with BPA to a final concentration of 0.3 mM (from a 0.3 M in 1 M NaOH freshly prepared stock) and cultured to mid-exponential phase (OD ~ 1). Cells were harvested by a centrifugation for 2 min at 3000 g and resuspended in 2 mL of cold water. Half of cells were transferred to a 12 well plate and subjected to UV irradiation for 1 h at 4 °C using a B-100AP lamp (UVP, CA). The other half of the cells was incubated on ice and served as non-irradiated control. After UV irradiation, cells were harvested by centrifuge spin for 2 min at 3000 g. Both irradiated and control cells were lysed in Lysis buffer (50 mM Tris.HCl [pH7.4], 200 mM NaCl, 1 mM EDTA, 1 mM PMSF (Roche) and 1x cOmplete protease inhibitor cocktail (Roche)by 5–6 × 1 min cycles of bead beating. Lysates were cleared by a 10 min centrifugation at 600 g. Cleared lysates were centrifuged at 100,000 g (25 min at 4 °C) in an Optima Max Tabletop Ultracentrifuge in a TLA 45 rotor (Beckman Coulter) to obtain crude membrane fractions. The membrane pellet was resuspended in denaturing buffer (50 mM Tris.HCl [pH7.4], 1 mM EDTA, 1% SDS, 2 M urea) and solubilized at 65 °C for 30–40 min with vigorous shaking. Insolubilized material was pelleted by centrifugation (15 min, 13000 g). The solubilized material was diluted with lysis buffer supplemented with 1% Nonidet P-40 and incubated overnight with anti-FLAG M2 magnetic beads—m8823 (Sigma-Aldrich). Beads were washed 3 times with lysis buffer/1% Nonidet P-40 and bound proteins eluted with SDS buffer and analyzed by immunoblotting.

**Native Immunoprecipitation**. Cells were in SD until med-exponential phase (OD ~1). Cells corresponding to 50 OD were then harvested by centrifugation at 3000 g and washed with lysis buffer (50 mM Tris.HCl [pH7.4], 200 mM NaCl, 1 mM EDTA, 2 mM phenylmethylsulfonyl fluoride and 1× cOmplete protease inhibitor cocktail). Lysates and crude membrane fractions were prepared as described above. Detergent extracts were prepared by solubilizing crude membrane fractions in lysis buffer/1% decyl maltose neopentyl glycol (DMNG). Insolubilized material was cleared by centrifugation (20,000 g, 15 min). The cleared detergent extracts were incubated overnight at 4 °C with FLAG M2 magnetic beads–m8823 (Sigma-Aldrich). Beads were washed 3 times with lysis buffer/1% DMNG, eluted with SDS-PAGE sample buffer and analyzed by immunoblotting. The input corresponds to 10% of the total extract used for IP.

**Gel electrophoresis and immunoblotting**. For protein quantification of whole cell lysates, cells were lysed using NaOH as described previously[70]. Briefly, cells pellets

corresponding to 1 OD were suspended in 0.15 M NaOH and incubated on ice for 10 min. Cells were pelleted, and resuspended in Laemmli sample buffer[71] and incubated 65 °C for 10 min with vigorous shaking. Debris was pelleted by a short spin, and samples were loaded on a 4–20% gradient SDS-polyacrylamide gel, separated by electrophoresis and blotted to a PVDF membrane.

**Fluorescence microscopy analysis of LDs**. For microscopy analysis of LDs, cells were in synthetic glucose media without inositol to stationary phase. Strains were inoculated at OD 0.1 and cultured for 20–24 h. Lipid droplets were visualized with BODIPY 493/503 (1 μg/mL).

Super resolution fluorescence microscopy was performed on an Olympus IX-83 inverted frame confocal microscope, equipped with a Yokogawa CSU-W1 SoRa super-resolution spinning disc module, and a Photometrics Prime BSI camera. Images were acquired using an UplanApo 60× objective (N.A. 1.50). Total magnification was 192×. BODIPY was excited using a 488 nm solid state laser (OBIS) at 5% intensity, and fluorescence emission was selected using a 525/50 nm bandpass filter. Images were processed for super resolution and deconvoluted (constrained iterative, maximum likelihood, 5 iterations) using Olympus cellSens Dimension software (version 3.1.1, build 21264). Figure preparation and quantification of LD size was done using ImageJ (version 1.53c; National Institutes of Health, USA).

**Lipid droplet biogenesis assay**. For induction of LD formation, strains were cultures in synthetic defined raffinose media (as SD but with 2% raffinose as carbon source). Overnight pre-cultures were inoculated at OD 0.25 and cultured for 3 h. Galactose was added to a final concentration of 2% to induce *DGA1* expression. LD formation was followed by staining with BODIPY and imaging by fluorescence microscopy.

Fluorescence microscopy was performed using a Zeiss Axio Observer.Z1 equipped with a Hamamatsu Orca Flash 4.0 digital CMOS camera. Images were acquired using an A Plan-APOCHROMAT 100× objective (N.A. 1.4). BODIPY fluorescence was analysed using a GFP-fluorescence set up consisting of a 485/20 bandpass excitation-filter (Zeiss), a 410/504/582/669-Di01 quad dichroic mirror, and a 525/30 bandpass emission filter. Microscope was controlled using Slidebook 6.0 software (3i).

**Statistics and reproducibility**. Statistical significance and *p*-values were calculated in GraphPad Prism 7 using the Kolmogorov–Smirnov test (2-tailed) or paired t-testing (2-tailed). Graphs were plotted in Prism. The following figure panels show representative data from at least three independent biological replicates that showed similar results: Figs. 3A, C, F, 4B, 5A, Extended Data Figs. S3D, G, H, S4A, B, C, S5A, C, E. The following figure panels show representative data from at least two independent biological replicates that showed similar results: Extended Data Fig. S3F and I.

**Reporting summary**. Further information on research design is available in the Nature Research Reporting Summary linked to this article.

## Data availability
Coordinates for the structures have been deposited in the Protein Data Bank under accession codes PDB 7OXP (Sei1) and 7OXR (Sei1ΔLH). The electron microscopy volumes have been deposited in the Electron Microscopy Data Bank under accession codes EMD-13103 (Sei1) and EMD-13104 (Sei1ΔLH). The following protein structures were obtained from the Protein Data Bank - PDB 6MLU (fly seipin) and PDB 6DS5 (human Seipin).

## Code availability
All codes used in this study are listed below and freely available as detailed.
Gromacs 2019.4 - https://www.gromacs.org/
VMD 1.9.4 - https://www.ks.uiuc.edu/Research/vmd/
Matplotlib 3.4.2 - https://numpy.org/
NumPy 1.16.2 - https://www.mdanalysis.org/

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

## Acknowledgements

We thank R. Klemm for critical reading of the manuscript and J. Ferreira for discussions. P.C. was supported by a BBSRC grant (BB/R018375/1) and an investigator award from Wellcome (202642/Z/16/Z). All electron microscopy data collection and processing were performed at the Central Oxford Structural Microscopy and Imaging Centre (COSMIC), which is supported by the Wellcome Trust (201536), the EPA Cephalosporin Trust, and a Royal Society/Wolfson Foundation Laboratory Refurbishment Grant (WL160052). S.M.L.'s lab was supported by a Wellcome Trust award (219477/Z/19/Z). R.A.C. and P.J.S. are supported by Wellcome (208361/Z/17/Z). P.J.S.'s lab is supported by awards from the BBSRC (BB/P01948X/1, BB/R002517/1, and BB/S003339/1) and MRC (MR/S009213/1). Simulations were performed using the ARCHER/ARCHER2 UK National Supercomputing Service (http://www.archer.ac.uk) and JADE, provided by HECBioSim, the UK High End Computing Consortium for Biomolecular Simulation (hecbiosim.ac.uk), which is supported by the EPSRC (EP/L000253/1). P.J.S. acknowledges the University of Warwick Scientific Computing Research Technology Platform for computational access.

## Author contributions

Y.A.K. carried out all the biochemical work and together with M.F.R. carried out light microscopy acquisition and data analysis. J.C.D. prepared the cryo-EM grids, collected and processed the EM data and determined the structure guided by S.M.L. R.A.C. performed the molecular dynamics simulations guided by P.J.S. P.C. conceived and guided the project and together with Y.A.K. and M.F.R. wrote the initial draft of the manuscript. All authors commented on drafts of the manuscript.

## Competing interests

The authors declare no competing interests
