## [Peer Review File · Nature Communications]

Mechanism of lipid droplet formation by the yeast Sei1/Ldb16
Seipin complexREVIEWERS' COMMENTS

Reviewer #2 (Remarks to the Author):

This study has been improved and I favor publication. I suggest using the word "nucleation" instead of "formation" in the title because it is more specific. Seipin probably plays roles in multiple steps in lipid droplet maturation and this study focuses on the earliest step, nucleation that is probably driven by phase separation.

Reviewer #3 (Remarks to the Author):

In this revised manuscript, the authors have properly addressed my concerns, mostly unifying nomenclature, improving labels and clarifying experimental details. I therefore recommend the acceptance of the paper and having it published as soon as possible. I would like to reemphasize that the current work has extract all the key findings in the well solved structure. The unique features of the yeast Seipin offers important opportunity for mechanistic probing. Based on these features, the work has analyzed the essential activity of Seipin in a systematic manner. It is a pity that the previous journal decided not to proceed despite the positive comments from the reviewers. I believe that the paper lays critical foundation for the field of LD biology and will certainly stimulate interesting discussion.

Response to the reviewers

Reviewer #2

“This study has been improved and I favor publication. I suggest using the word “nucleation” instead of “formation” in the title because it is more specific. Seipin probably plays roles in multiple steps in lipid droplet maturation and this study focuses on the earliest step, nucleation that is probably driven by phase separation.”

We thank the reviewer for the suggestion. While our molecular dynamics experiments clearly implicate Seipin in the nucleation step, the aspects of Seipin function characterized in our study are likely to impact other steps of the lipid droplet process. Moreover, all the functional assays used in the paper monitor lipid droplet formation and not specifically the nucleation step. Therefore, we believe that the terminology used, while broader, is the most accurate.